# Social Factors as Major Determinants of Rural Development Variation for Predicting Epidemic Vulnerability: A Lesson for the Future

**DOI:** 10.3390/ijerph192113977

**Published:** 2022-10-27

**Authors:** Małgorzata Dudzińska, Marta Gwiaździńska-Goraj, Aleksandra Jezierska-Thöle

**Affiliations:** 1Institute of Spatial Management and Geography, Faculty of Geoengineering, University of Warmia and Mazury in Olsztyn, 10-719 Olsztyn, Poland; 2Institute of Geography, Kazimierz Wielki University, Plac Kościeleckich 8, 85-033 Bydgoszcz, Poland

**Keywords:** socio-economic geography, epidemic, COVID-19, environmental management, sustainable development, wellbeing, spatial planning, geographical information system

## Abstract

There have been changes in social attitudes in recent years. These changes have been a consequence of a new societal view of the common good, which manifests itself in social responsibility for a clean and healthy environment. The outbreak and spread of the COVID-19 epidemic has highlighted the socio-spatial variation across regions and countries. The epidemic necessitated restrictive measures by state authorities. In the initial period in many countries, the actions of the authorities were identical throughout the country. This was mainly due to a lack of information about the differentiation of areas in relation to the epidemic risk. The aim of the research was to present a model for classifying rural areas taking into account vulnerability to epidemic threats. The model takes into account demographic, social, economic and spatial-environmental development factors. A total of 33 indicators based on public statistics that can be used to determine the area’s vulnerability to epidemic threats were identified. The study showed that for Poland, 11 indicators are statistically significant to the developed classification model. The study found that social factors were vital in determining an area’s vulnerability to epidemic threats. We include factors such as average number of persons per one apartment, village centers (number), events (number), number of people per facility (cultural center, community center, club, community hall), residents of nursing homes per 1000 inhabitants, and the number of children in pre-school education establishments per 1000 children aged 3–5 years. The research area was rural areas in Poland. The results of the classification and the methods used should be made available as a resource for crisis management. This will enable a better response to threats from other epidemics in the future, and will influence the remodeling of the environment and social behavior to reduce risks at this risk, which has a significant impact on sustainable development in rural areas.

## 1. Introduction

Sustainable area development has an impact on people’s quality of life directly as it affects people’s income, but also indirectly through the stimulation and activation of economic activity, the development of social and technical infrastructure and the improvement or deterioration of the environment. In the literature, there are various definitions of well-being, which can result, among other things, from lifestyle, the ability to meet life’s needs, but also be seen through a sense of health. Health issues are considered as an important part of the sense of well-being, which was clearly noted with the COVID-19 outbreak. It is therefore important to present a model for classifying rural areas taking into account vulnerability to epidemic threats.

The outbreak, rate and range of spread of the COVID-19 epidemic gave rise to scientific works in many fields—medicine [1,2], social sciences [3,4], and economics [5]—and surveys related to both local area [6], regions, countries and continents, e.g., Asia [7], Africa [8], Europe [9] and America [10]. Previously, many studies were devoted to the spread of epidemics in space and time [11]. Epidemiological risk is mostly associated with infectious diseases and preventing their occurrence and spread [12]. Most dangerous diseases pestering humanity for centuries were gradually contained, and in the second half of the 20th century attention was paid to the spread of infectious diseases. The cause-and-effect relationship between the frequency of social (interpersonal) relations and the intensity of infections was demonstrated [13]. More than 2000 years ago Hippocrates claimed that environmental factors could lead to diseases. However, no measures were taken until the 19th century when intensive investigation into phenomena determining the spread of diseases in human populations commenced [14]. Studies comparing prevalence rates of human diseases became an important tool for demonstrating a relationship between living conditions or environmental factors and certain diseases. This approach was mainly used in epidemiology [15], where the subject of study is the population living in a specific area at a specific time. The population structure in various areas differs and varies in time. In epidemiological analyses, behavior, lifestyle and social relations affecting human health are essential [16].

Meanwhile, the outbreak and spread of the COVID-19 epidemic highlighted the socio-spatial differentiation of respective regions and countries [17]. The epidemic has revealed spatial differentiation associated with patterns of interpersonal relations and social interactions, as COVID-19 is a droplet-borne virus spreading faster through proximity and direct social interaction [18]. The characteristics of a society reveal how people interact and thereby spread diseases [19]. Their significance may vary in respective countries due to tradition, mentality and lifestyle. Many cause-and-effect relationships affect interpersonal and social interaction patterns. They include are interactions between groups or between an individual and a group, manifested as mutual influence on their behavior, sometimes referred to as social activity [20,21]. Social interactions result from family, social and professional ties and can be looked into from the perspective of their direction and intensity, form and structure. They can also be investigated in terms of the underlying demographic, social, and economic factors specific to a local community. The structure of social interactions relates to the configuration of possible links between entities. These can be arrangements between two persons, groups, or individuals and groups as well as associations, networks, coalitions and clusters. The direction, intensity and form of social interaction are interlinked, constituting a complex process which—due to the human factor—can take a different course even with the same participants. In daily life, a person enters into social interactions intentionally—consciously seeking them and targeting their actions at other people [22,23]—or unintentionally—when in certain relations individuals or groups of people interact accidentally [24]. The first—intended—relation may relate to establishing and maintaining family or professional interactions in connection with needs, objectives and motives, with a permanent frequency. The other—unintended—relation refers to actions aiming to limit the influence of others; encounters and contacts with other people are occasional and do not aim at maintaining ties. Due to globalization and migration, infectious diseases originating in distant countries can spread across countries in a short time, thus setting a new direction for epidemiological transition [25,26].

Scientists are vying to identify such relationships around the world [24] and in their countries [27]. An active debate is pending on critical socio-economic determinants of the epidemic’s spread. Experts claim that countries most affected by the COVID-19 pandemic feature a high percentage of seniors (+65) [28,29] or poorly developed healthcare systems [30,31].

Apart from biological and epidemiological factors, multiple social and economic criteria determine the spread of COVID-19 in a population [32,33]. Others underline the role of natural environment factors [32,34]. Physical and geodetic space and relations between its elements are essential [35].

Government policy—social distancing, testing, and restrictions imposed and loosened as the epidemic weakens—is also significant. These measures significantly contribute to maintaining labor market stability and economic growth. Considering that in the past decades Europe was not affected by epidemics and medicine focused on diseases of contemporary civilization, the public health vigilance and prevention of infectious diseases were impaired [36]. To satisfy the healthcare needs of the population, every country should have an efficient epidemic management system in place, oriented both at improving human health and preventing the spread of pandemics related to social interactions. Crisis management efficiency at the national, regional and local level is determined by up-to-date reliable knowledge of social interactions.

Therefore, this study attempted to present a model for classifying rural areas considered as vulnerable to epidemic threats transmitted via the droplet route. It took into account demographic, social, economic and spatial development factors indirectly reflecting the level of interpersonal relations and social interaction. Previous studies deemed them as determinants of the spread of pandemics, including COVID-19 [37,38], with new elements [39]. The factors were also assigned—according to types of interpersonal and social relations—to two groups: intended and unintended interactions.

Eighty-three studies on establishing the relationship between socioeconomic and spatial development factors and the susceptibility of a given area to epidemic threats were revised using the Scopus, Web of Science Core Collection and Google Scholar browsers. The keywords used included COVID-19, social factors, economic factors, space and spatial development, epidemic threats, and geographical concentration. Based on the review of the articles, a total of 33 factors were identified (Table 1). The preliminary literature review found that the COVID-19 epidemic developed faster in economically highly developed areas with higher a GDP [17]. Other factors correlated with the number of cases and deaths from COVID-19 are the average age, population density and the share of people employed in caring for the elderly, the percentage of school-age children and children in daycare, and the density of doctors [32]. The spread of COVID-19 has also been linked to people travelling (migrating) from Wuhan in the Hubei province [4] and migration in the country based on returning tourists, mainly from ski resorts [33]. Another accelerator of the pandemic was participation in festivals [33].

Previous studies into the spread of COVID-19, MERS-CoV, SARS-CoV, and Ebola viruses showed that socio-economic conditions should be considered in creating policies regarding the distribution of resources at the beginning of the epidemic. Knowledge of the factors that help to determine how vulnerable an area is to epidemic threats and thus affect the frequency of diseases in populations is essential to national, regional and local authorities dealing with healthcare. It can improve threat identification and contribute to the correct use of resources and adequate preventive schemes. Understanding the mechanisms behind the spread of epidemics is essential for formulating correct optimum preventive schemes.

The preliminary review of reference literature led to the following research hypothesis: socio-economic factors and spatial development can determine the scale, intensity and concentration of areas with different levels of vulnerability to epidemic threats.

The model for classifying areas with different levels of vulnerability to epidemic threats hereinafter referred to as the model for classifying areas with different levels of vulnerability to epidemic threats (KLE) would ensure the optimum implementation of prevention and control of the epidemic spread in emergencies, and can also be a helpful instrument in determining directions for modeling the environment and social behavior to reduce the risk of an epidemic in the area, which has a significant impact on the sustainable development of rural areas.

To the authors’ knowledge, this topic has not been studied in the literature, so there is a need to fill the gap in the literature and practice regarding the problem addressed.

This paper is innovative for three main reasons: (1) it comprehensively identified and classified factors available from public statistics that allow the classification of areas vulnerable to epidemic threats; (2) the significance level of the proposed factors in determining the susceptibility to epidemic threats for the territory of Poland was determined; (3) the level of susceptibility of rural areas of Poland to epidemic threats was determined.

These findings can be used for developing a specifically oriented policy and planning resources in case of a future pandemic caused by a similar virus. The research problem raised in connection with modeling areas vulnerable to epidemic threats and the spread of COVID-19 extends the scope of previous research based in Italy [40], England [41], Germany [33], India [42], Pakistan [43], Poland [44], China [45] and the USA [46,47]. The results can provide essential knowledge to allow fast, relevant and efficient preventive and protective action. The article is divided into five sections. In the Section 1, the introduction describes the theoretical framework related to the vulnerability of areas to epidemic threats, while the Section 2 presents the proposed research methodology. The Section 3 describes the results of empirical research in which a preliminary model of vulnerability to epidemic threats is presented. The Section 4 contains the discussion, while the Section 5 covers conclusions and limitations. The paper also contains the authors’ own reflections and general recommendations.

## 2. Materials and Methods

### 2.1. Materials

The study area was Poland, located in Central Europe, on the northern and eastern hemisphere, in middle latitudes, between 49°00′ and 54°50′ north latitude and between 14°07′ and 24°09′ east longitude. The administrative area of Poland is 312,679 km^2^. According to Statistics Poland, in 2018, rural land in Poland accounted for 93% of the country’s overall area. The study area is located in the European Union and comprises 314 rural units in Poland (Local Administrative Units LAU—1 local level) (Figure 1).

### 2.2. Methods

The work attempted to illustrate a model for classifying areas with different levels of vulnerability to epidemic threats (KLE), taking into account demographic, social, economic and spatial development factors as well as types of intended and unintended interpersonal and social relations. The model would facilitate the optimum implementation of measures to protect society in the event of epidemic emergencies. The pattern presents the course of the research process (Figure 2).

The stages of research were:

Stage 1. Identifying socio-economic and spatial development factors indirectly reflecting the level of interpersonal relations and social interactions

The factors indirectly reflecting the level of interpersonal relations and social interactions were presented in two aspects—according to their nature (demographic, social and economic, spatial development) and type of interpersonal and social relations (intended and unintended). Diagnostic factors were selected based on a thorough review of reference literature (see Table 1).

The factors were substantially associated with the modeled phenomena, because the variables should be good representatives of the analyzed groups of factors. The mutual relationship between the variables should be logical; the factors should have a defined research tradition, which implies a pre-established interpretation and substantive control; they should be accessible and reliable; the characteristics should be (directly or indirectly) measurable. Those that are not directly quantifiable should be transformed into measurable characteristics [48]. Considering that the rate at which infectious disease (including COVID-19) spread is affected by specific features of the dwelling place, demographic characteristics of the community, occupation, and economic development of the region, a set of different factors was identified, including demographic, social, economic and spatial development factors. In addition, attention was paid to the type of social relations (e.g., relations at work), which can be intended or unintended (e.g., meeting at the shop). It should be noted that the selection and definition of factors was partly limited by the availability of official statistics [49,50]. The surveys took place at the district level (LAU 1 in the Polish statistical system) including rural districts.

The variables were based on statistics from the Local Data Bank of Statistics Poland (https://bdl.stat.gov.pl/, accessed on 29 December 2020) (see Table 1). The time range for the diagnostic factors covered the latest data for 2019—prior to the outbreak of the COVID-19 pandemic. Such an approach made it possible to identify and describe the dynamics of spatial changes not connected with restrictions imposed due to the pandemic. Next, quasi-variables, i.e., not contributing new information, were eliminated. To this end, the variation coefficient was calculated for each factor, eliminating the features for which it did not exceed 0.1. Afterwards, the Pearson correlation coefficient was used to measure the strength of the relationship between other variables. Its critical value is above |0.6| [51].
ijerph-19-13977-t001_Table 1Table 1Diagnostic factors for classifying areas vulnerable to epidemic threats.FactorSocial InteractionsType of FactorFactor SelectionLiterature SubjectNature of the FactorsType of Interpersonaland Social RelationsDSLEIUX1—population per 1 km^2^x


xxdestimulantaccept[32,52]X2—share of population aged 6–19 years and more (%)x


x
destimulantaccept[53]X3—share of population aged 60 and over (%)x


xxdestimulantreject[32]X4—population per townshipx



xdestimulantreject[32,52]X5—balance of internal migration per 1000 populationx


xxdestimulantaccept[17,32]X6—balance of foreign migration per 1000 populationx


xxdestimulantaccept[17]X7—registered unemployment rate (%)
x


xstimulantaccept[17,32,52]X8—average number of persons per 1 apartment
x

x
destimulantreject[32,52]X9—average number of persons per 1 room
x

x
destimulantaccept[32,52]X10—number of accommodation establishments over 10 people


xxxdestimulantreject[17,32]X11—number of vehicles per 1000 population


xx
destimulantaccept[32,52]X12—hypermarkets (number)


x
xdestimulantaccept[54]X13—supermarkets (number)


x
xdestimulantreject[54]X14—marketplaces (number)


x
xdestimulantaccept[54]X15—village centers (number)
x

x
destimulantreject[54]X16—events (number)
x

x
destimulantreject[55]X17—number of event attendees per 1000 population
x

x
destimulantaccept[55]X18—number of people per facility (cultural center, community center, club, community hall)
x

x
destimulantaccept[55]X19—number of beds in sanatoria
x

x
destimulantreject[54]X20—population density in housing areas (person/1 km^2^) *

x

xdestimulantaccept[56]X21—population density of built-up and urbanized areas (person/km^2^) *

x

xdestimulantaccept[56,57]X22—average useful floor area of a dwelling per 1 person
x

x
stymulantreject[57,58]X23—population density of industrial areas (person/1 km^2^) *

x
xxdestimulantaccept[17,57,58]X24—number of towns

x

xdestimulantaccept[58]X25—business entities by size classes per 1000 inhabitants in total


xxxdestimulantaccept[17]X26—ambulatory health care—medical consultations per 1000 population
x


xdestimulantReject[59]X27—physicians (total working staff) per 10,000 population
x


xdestimulantaccept[59]X28—residents of nursing homes per 1000 inhabitants
x

x
destimulantaccept[60]X29—children in pre-school education establishments per 1000 children aged 3–5 years
x

x
destimulantaccept[61]X30—national economy entities employing more than 49 persons per 10 thousand population


xxxdestimulantreject[17]X31—number of bed places per 1000 population


xxxdestimulantreject[17,32]X32—tourists using accommodation per 1000 population


xx
destimulantaccept[17,32]X33—% degree of utilisation of accommodation%


xxxdestimulantaccept[17,32]D—demographic, S—social, E—economic, L—land use, I—intended, U—unintended. * data are from 2014. Source: www.stat.gov.pl (accessed on 29 December 2020).

Stage 2. Developing and evaluating the model for classifying areas with different levels of vulnerability to epidemic threats (KLE) and component models.

The statistical procedure eliminated certain indices adopted for the KLE model and a set of 21 diagnostic indices classifiable in terms of the nature of factors and type of relationship was designed. The diagnostic indicators included both stimulants and destimuants. Diagnostic indicators were expressed in different units. Therefore, these indicators were normalized and adjusted for comparability by removing the appropriate measurement units and standardizing all variables by transforming them into stimulants. It was done using the zero unitarization method, where the following transformation operations were applied:

For stimulants:$$v_{ij}=\frac{x_{ij}-minx_{ij}}{maxx_{ij}-minx_{ij}}$$

For destimulants:$$v_{ij}=\frac{maxx_{ij}-x_{ij}}{maxx_{ij}-minx_{ij}}$$
where:

*v_ij_*—standardized value of the indicator *x_ij,_*

*x_ij_*—value of the *j*th diagnostic indicator of an *i*th object

*minx_ij_*—minimum value of the *j*th diagnostic indicator *x_ij,_*

*maxx_ij_*—maximum value of the *j*th diagnostic indicator *x*.

The indices were used for computing a general synthetic index (WO) and component indices grouped diagnostic factors according to their nature: demographic (Wd), social (Ws), economic (Wg), spatial development (Wzp), and according to the type of interpersonal and social relations: intended (Wz), unintended (Wn). Two separate classifications were designed (Figure 2).

The reference literature offers a wide range of aggregation methods [62,63]. Most frequently, additive methods are used—from sums of unit ranks for each index (equal weight) to aggregation of weighted transformation of original indices (expert weight) [64]. The standardized sums method (the Perkal index) was used according to the following formula.
$$w=\frac{\sum_{j=n}^{n}V_{ij}}{n}$$
where:

*w*—synthetic indices (WO, component indices: Wd, Ws, Wg, Wzp, Wz, Wn)

*vij*—standardized index value in i-th case and for j-th variable

*n*—number of features analyzed.

As a result, the diagnostic indices were transformed accordingly into synthetic indices: WO (general), Wd, Ws, Wg, Wzp, Wz, Wn (component).

The next stage was classification of rural districts (LAU-1) that are vulnerable to epidemic threats. Four classes (I–IV) were defined, where districts from one group had a similar level of vulnerability to epidemic threats. The grouping used the arithmetic mean of synthetic ranks for all districts (*Rav*) and standard deviation (*s*), assuming that districts featured the following levels of vulnerability to epidemics: low, in which to be classified into this group, the counties had to meet the following condition: (*Ri* > *Rav* + *s*); medium (*Rav* + *s* > *Ri* > *Rav*); quite high (*Rav* > *Ri* > *Rav* − *s*); high (*Rav* − *s* > *Ri*).

The KLE model was based on the computed general synthetic index (WO), and component models—on component indices grouping diagnostic factors (Figure 2).

Stage 3. Identifying the relationship between the course of the COVID-19 epidemic and the KLE model and component models.

The course of the epidemic was established based on the number of registered COVID-19 infections. To identify the relationship, the identification of the spatial diffusion of registered COVID-19 infections in Poland in the analyzed LAU-1 districts was necessary. To this end, the intensity of detected COVID-19 infections was illustrated at intervals of 30 days between 30 April 2020 and 30 November 2020. The survey used statistics compiled pro bono by the team of Michał Rogalski (30). The data were updated day by day with information provided by regional and district sanitary–epidemiological stations until 23 November 2020. From 23 November infection statistics have been posted on the Polish government’s website at https://www.gov.pl/web/koronawirus/wykaz-zarazen-koronawirusem-SARS-COV-2 (accessed on 15 March 2021).

The term of study covered the first pandemic wave in Poland when vaccines were not available. The relationship between the course of the epidemic and the KLE model and component models was identified using the Pearson correlation coefficient. Correlations were established between the general synthetic index (Wo), KLE model and component indices and the intensity of detected COVID-19 infections. This verified the correctness of the KLE model.

Stage 4. For the final selection of the attributes for the KLE model, we used a stepwise regression model [65]. Determination of the linear multivariate stepwise regression model was performed for the selection of attributes. Stepwise regression selects the explanatory variables for multiple regression models based on their statistical significance.
$$Y=a_{0}+a_{1}\times x_{1}+\cdots +a_{i}\times x_{i}$$
where:

*Y*—dependent variable (infections)

*X*_1_—independent variable

*a*_0_—free term

*a*_1_,*a*_i_—model parameters, coefficients determined with the least square method.

Stage 5. Determining the model of vulnerability to epidemic threats—improved KLE model based on features with statistical significance.

Stage 6. Identifying relationships between the attributes for the KLE (Stage 4), and improved KLE model (Stage 5). We determined a linear model with one variable.
$$x_{1,2,\ldots ,n}=b_{0}+b_{1}\times Y$$
where:

*X*_1,2,…, *n*_—attributes for the KLE

*Y*—improved KLE model

*b*_0_—free term

*b*_1_—model parameter.

## 3. Results of Empirical Surveys

Delimitation and evaluation of the model for classifying areas with different levels of vulnerability to epidemic threats (KLE) and component models.

The general synthetic index (WO) and component indices were used to evaluate the spatial differentiation of rural areas in Poland in terms of vulnerability to epidemic threats. The models assumed that the lower the synthetic index is, the higher the class of the area and thus the higher the likelihood of interpersonal relations and social interactions, with more favorable factors occurring due to the functional structure of the space.

In the model (KLE), areas situated near cities constituting centers of regions and in the south of Poland (Lesser Poland region) are the most vulnerable to epidemic threats. On the other hand, areas at the north-east and east border are the least vulnerable to epidemic threats (Figure 3). The spatial development level considerably influences this situation, as presented in the discussion of the component models.

The component models—demographic (Md), social (Ms), economic (Me) and spatial development index (Isd)—feature very different spatial arrangements, implying a different impact of respective factors on vulnerability to epidemic threats (Figure 4).

In the demographic model (Md), the highest class (IV) was noted in units adjoining the capitals of regions that are attractive places to live in. In terms of demography, they feature high density and inflow of the population and a favorable share of young inhabitants. The second group comprises units scattered in the regions of Pomerania, Greater Poland, Lesser Poland and Masovia. The lowest vulnerability—class (I) was noted mostly in districts featuring low densities and an outflow of populations, as well as a low share of young people under 19.

The social model shows mosaic-like spatial arrangements. The concentration of class IV districts is clear in Greater Poland and Lesser Poland due to labor market factors (unemployment rates) and the location of social infrastructure, including housing conditions.

Highly vulnerable areas (IV) in the economic model (Me) are mainly cities and large concentrations of tourist services.

The spatial development index (Isd)—mostly coinciding with the existing links with capitals of regions within the settlement network and solutions resulting from industrial development—is essential to determining the vulnerability of areas to epidemic threats. The impact zones of the urban–rural continuum of Warsaw, Kraków, Poznań, Katowice, Gdańsk and Wrocław, where the highest values of the index were recorded, are clearly marked.

High spatial differentiation of rural districts (LAU 1) was found for synthetic indices describing the type of interpersonal relations (intended and unintended) (Figure 4). The spatial distribution of districts, illustrating the effect of spatial development and the unintended interpersonal relation factors, coincided to a high degree. On the other hand, the spatial distribution of districts illustrating the effect of demographic factors was similar to the distribution of the intended interpersonal relations factor.

Figure 5 presents the spatial distribution of component models describing the type of interpersonal relations—intended and unintended. The intensity of intended relations (Ri) was highly associated with the dwelling place and employment and dominated in suburban rural areas with dominant economic functions, where the locations of businesses and tourist development intensity were significant (Figure 5). On the other hand, the intensity of unintended relations varied due to the distribution of the population and social infrastructure accessibility. Next to urbanized rural areas, near the capitals of voivodeships and central cities, areas associated with cross-border traffic with Germany (Szczecin, Zgorzelec), in the port city of Gdańsk, and with Ukraine (Przemyśl) clearly stand out. Tourist traffic in the Carpathians and Sudetes area is also significant.

## 4. Discussion

Satisfaction of the healthcare needs of society is essential to developing national healthcare policy and should be supported by efficient crisis management at the national, regional and local level. Health issues are an important part of wellbeing, which refer to an individual’s subjective sense of satisfaction with psychological, physical and social living conditions. Our well-being also depends on the environment in which we live, work and rest. This is why it seems so important to understand what factors influenced the spread of the COVID-19 outbreak. Understanding the impact of external factors is particularly important in the context of sustainable development. Specific knowledge derives from epidemiological, demographic, economic, environmental and spatial studies (39). Research in this area is extremely important as it allows appropriate decisions to be taken in epidemiological management.

### 4.1. Strength of the Relationship between the Classification of Areas in the KLE Model and the Spread of COVID-19

The strength of the relationship was determined between the model for classifying areas with different levels of vulnerability to epidemic threats (KLE) and the COVID-19 spread pattern (registered cases). Presentation of this relationship requires showing the stages of COVID-19 virus spread in Poland in the spatial distribution of LAU-1. The number of COVID-19 cases in Poland varies between administrative regions. Figure 5 shows the number of registered COVID-19 infections in the analyzed districts in Poland from 30 April 2020 to 30 November 2020. The spatial analysis clearly implies certain relationships. At the onset of the epidemic, the southern, most industrial part of Poland where coal mines and big factories are situated was the most affected. Later, the disease spread in the south–north and west–east direction. Firstly, the spread to the north resulted from the migration of people working abroad. Secondly, the spread from cities to rural areas was caused by rural inhabitants commuting to work in the city. A strong concentration was noted in the region of Masovia with Warsaw—the capital of Poland—and in Silesia and Lesser Poland (dense building development, fragmented agricultural land, industrial zones). Initially (04–07.2020), COVID-19 spread slowly, but in summer the number of infections grew, giving rise to a dynamic increase in the last analyzed period (30 November 2020) and next stage of the pandemic (Figure 6).

Summing up, after [16], mobility at a supraregional and international level and to conurbations [35,66]—manifested in commuting from rural areas to the city and abroad—and tourist traffic was very important to the spread of the epidemic. Seasonal workers are usually employed for a specified period of time and live in worse conditions, in overcrowded districts and suburbs, so the risk of infection is higher. When they return home, they infect their families [67].

Analyzing the spatial distribution obtained, it was found that in the initial phase of the pandemic, no significant relationship between the KLE model and the spread of COVID-19 occurred. However, with time, as the incidence increased, the relationship between these variables gradually increased. A clear upward trend was observed for the correlation in time (Figure 7). It was high in October 2020 (−0.619) and November 2020 (−0.669). Thus, in this interval, a high negative correlation was recorded between the classification of rural areas vulnerable to epidemic threats and the number of COVID-19 infections in Poland. This fact clearly shows that the spread of COVID-19 was influenced by the following: population density, the share of young people under 19, population density in housing and industrial areas, internal and external migrations, etc.

The time shift in the relationship between the proposed classification of districts at a regional level and the number of registered infections may be due to four reasons.

Firstly—imperfect registration of COVID-19 cases. [35] found that detectability of the disease is determined by its correct diagnosis in the population. With COVID-19, this all depends on testing. In Poland, testing took place primarily in places where exposure occurred more frequently—in health and community care centers (e.g., hospitals, nursing homes) and in communities (families, workplaces and other concentrations of people) with already-identified infections. Thus, data on infections are not representative of the whole infected population. This is due to the disease itself—as noted, some infected people show no symptoms. Based on screening in various parts of Poland (Kraków, Upper Silesia; [68]), supposedly only some cases in Poland are detected.

The second reason is state interventionism, the extent of measures undertaken by the state, local authorities, etc., to limit the spread of the virus and the disease [69]. They aimed to considerably restrict interpersonal relations and social interactions through isolation or quarantine orders [70]. Many studies evaluate government interventionism in combating the pandemic. Different researchers, including [71,72], examined the effect of various orders on the virus spread level. SAH restrictions raised intense political debate [73]. The results of surveys are partly consistent and show that the measures decreased the transmission of the epidemic. State interventionism slowed down the epidemic in Poland in the first half-year of 2020 and was identical in all Poland, without regional variations.

The third reason is the characteristics of the SARS-CoV-2 virus, including its nature and spreading route. SARS-CoV-2 (Severe Acute Respiratory Syndrome Coronavirus 2) is highly contagious and leads to severe acute respiratory distress, a disease known as COVID-19 (Coronavirus Disease-19). It is transmitted via droplets, which is specific to this virus strain. Not everyone coming into contact with this virus gets infected. So far it has not been established why a part of the population is immune or shows no symptoms. Since viruses having genetic material recorded in RNA thread are more variable than DNA viruses, it can be expected that “new viruses” will occur [74,75].

The fourth reason was social behavior and compliance with existing orders and restrictions. At the beginning, Poles and their families were suddenly faced with strong restrictions and the media was overflowing with catastrophic news from the most affected countries such as Italy and Spain [76]. This contributed to a change in the behavior and mobility of Poles at that time.

After Czech (2020), changes in social behavior were presented based on data about the mobility of Poles from Google’s COVID-19 Community Mobility Reports (Figure 8). In the first months of the pandemic, people stopped travelling to places associated with entertainment, culture and relaxation (S1 category) and ceased using public transport (S3). Additionally, less people commuted to work (S4) or went to grocery stores and pharmacies (S2). The outbreak of the COVID-19 pandemic resulted in the closing of schools and orders to work from home (S5). Starting from April 2020, in Poland restrictions related to COVID-19 were waived, which reversed trends for all the analyzed mobility categories. Places associated with trade in the base period before the pandemic have gradually regained their popularity. However, public transport was less preferred and a part of the population still worked from home (as of 14 June 2020) [76]. Holidays intensified the mobility of Poles who—after several months of restrictions—travelled en masse to places associated with entertainment, culture and relaxation (S1), which increased the number of infections in subsequent months (October, November 2020).

The thematic map (Figure 9) illustrates the spatial relationship between the KLE model and the number of registered cases of COVID-19. The spatial analysis of the relationship clearly implies that at the onset of the epidemic (30 April 2020), the highest dependency (high WO and prevalence) in Poland was noted primarily in rural areas near urbanized and industrial zones adjacent to big cities. This relationship reflects the functional and spatial structure [39]. In November 2020 (end of the first epidemic wave) the disease moved to the remaining rural areas and a clear correlation was noted between the classification of areas according to their vulnerability to epidemic threat and the number of cases. Many infections were recorded near Warsaw, Poznań and in Lesser Poland.

### 4.2. Correlation between Component Models and the COVID-19 Spread Pattern

The Pearson correlation was carried out between component models and the COVID-19 spread pattern (Table 2).

A high correlation was identified in November 2020 with the demographic (Md) and spatial development (Isd) model and the model illustrating the type of unintended relations and it was −0.683 for the latter. This shows that features due to functional and spatial structure are significant to the spread of COVID-19.

Selection of attributes for the KLE model (pop)—stepwise regression model.

The estimation of regression parameters was carried out by the least squares’ method. Based on the results, we find that the estimated model explains more than 77% of the variability of the original dependent variable. As the F test for eleven independent variables and 298 cases was F = 86.136, the hypothesis that the regression coefficients were not statistically significant was rejected and an alternative hypothesis was adopted. The values of the t-statistic indicate that the intercept and regression coefficients are also significantly different from zero.

The obtained analysis shows that 11 variables were statistically significant (Table 3).

### 4.3. Revised Model of Vulnerability to Epidemic Threats—Poland

A prerequisite for the formulation of programs to facilitate the prevention of an epidemic in the territory of the country is to identify areas with features of appropriate sensitivity to this threat. Ultimately, for the territory of Poland, a WO(pop) susceptibility model was determined based on 11 variables. The obtained classifications are presented in the figure. In the obtained model, the greatest number of variables is from the group of social factors. We included factors such as X8—average number of persons per 1 apartment, X15—village centers (number), X16—events (number), X18—number of people per facility (cultural center, community center, club, community hall), X28—residents of nursing homes per 1000 inhabitants, X29—children in pre-school education establishments per 1000 children aged 3–5 years.

In the next stage (6) of the research, the relationship between the 11 finally chosen variables for the model was determined [41] and the revised WO(pop) model (Figure 10) was created. The relationships between the KLE-independent variable and the dependent variables are shown in the two-dimensional scatter plots in Figure 11. In each scatter plot, points represent individual LAUs. The goodness-of-fit of the model was determined on the basis of the calculated value of r^2^, which showed that the variable X13 (supermarkets (number)) is a moderate-level variable correlated [51] with the improved model WO(pop), and the variables X1, X5, X16, X21, X29 with a low level of correlation [51]. The model can be interpreted as follows:
If the value of the variable X13 (supermarkets (number)) increases by one unit, the value of the WO(pop) composite index increases by 0.0065.If the value of the variable X1 (population per 1 km^2^) increases by one unit, the value of the composite index WO(pop) increases by 0.0467 (Figure 11).If the value of the variable X5 (balance of internal migration per 1000 population) increases by one unit, the value of the composite WO(pop) index increases by 0.0027.If the value of the variable X16 (events (number)) increases by one unit, the value of the composite WO(pop) indicator will increase by 0.2236.If the value of the variable X21 (population density of built-up and urbanized areas (person/km^2^)) increases by one unit, the value of the composite WO(pop) index increases by 0.3182.If the value of the variable X29 (children in pre-school education establishments per 1000 children aged 3–5 years) increases by one unit, the value of the WO(pop) composite indicator will increase by 0.0515.

The change of the X16 feature from the group of social factors and X21 from the group of spatial-environmental factors have the greatest impact on the change of the WO(pop) index.

The results of the analysis have important implications for planning measures to prevent the spread of another epidemic, including the restructuring of the existing spatial structures in the context of demographic, social and economic changes. The analysis also showed which of the analyzed variables have a significant impact on the WO(pop) model.

## 5. Conclusions

The present study used geographic methods and tools to identify and classify the level of vulnerability of rural areas to epidemic threats. A total of 33 indicators based on public statistics that can be used to determine the area’s vulnerability to epidemic threats were identified. The study showed that for Poland, 11 indicators are statistically significant to the developed classification model. The study found that social factors were vital in determining an area’s vulnerability to epidemic threats. A stepwise regression model was used for the final selection of attributes for the rural area classification model.

The results of surveys corroborated the research hypothesis that socio-economic factors and spatial development can determine the scale, intensity and concentration of areas with different levels of vulnerability to epidemic threats. The proposed classification of rural areas with different levels of vulnerability to epidemic threats takes into account demographic, social, economic and spatial development factors. The results of research partly coincide with the results of other researchers [16], including English researchers such as Paul [46] and Italian ones who analyzed the relationship between types of rural landscape and the epidemic [40] and researchers from Asia [38], but were additionally an attempted new empirical approach to classify rural areas in terms of their vulnerability to epidemic threats.

The classification was evaluated and spatial patterns were verified by establishing how they related to the epidemic of COVID-19. The results show a close relationship between the classification and occurrence of COVID-19, which implies that the approach is correct. Given the plurality of factors (11 diagnostic indices), the results were soundly corroborated. The results of the classification of rural areas and the methods used should be made available as a crisis management resource. It is also an instrument that should be used to indicate the directions of remodeling the environment and social behavior in order to reduce the risk associated with this threat, which has a significant impact on the sustainable development of rural areas

This knowledge will allow a precise, improved response to future threats caused by other epidemics transmitted through interpersonal contact (droplets). The methods used and the model presented allows classifying the areas covered by a potential new epidemic wave in the future. Timely and efficient application of adequate government strategies would slow down the spread of the epidemic through limiting human mobility. Knowing how vulnerable an area is to epidemic threat will help improve and set the direction for action, i.e., lockdowns only in the most-exposed areas instead of whole regions, thus avoiding a complete paralysis of social and economic life.

The proposed methods and classification of rural areas are universal and can be applied to other countries. However, the factors must be aligned with national statistics. It will allow undertaking epidemiological management measures in other countries.

### Limitations

The study is not a COVID-19 diffusion study. Classifications of areas’ vulnerability to epidemic threats based on the prevalence of the COVID-19 virus were made. The future development trends of the new epidemic are uncertain. Other droplet-borne viruses may have different epidemic characteristics, and the relationship between the presented factors may differ.

Future research directions: it would be advisable to compare the area’s susceptibility to epidemic threats based on other epidemics and obtain a classification for the area at the commune or precinct level.

## Figures and Tables

**Figure 1 ijerph-19-13977-f001:**
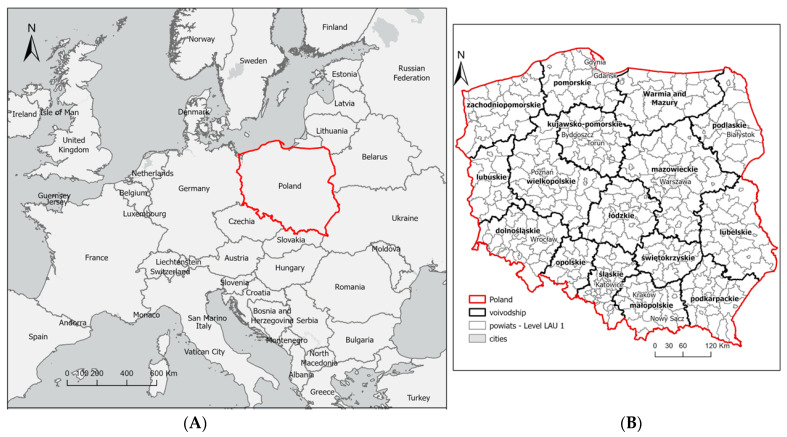
Location study area. (**A**) Location of Poland in Europe. (**B**) Level LAU 1. Source: Own elaboration.

**Figure 2 ijerph-19-13977-f002:**
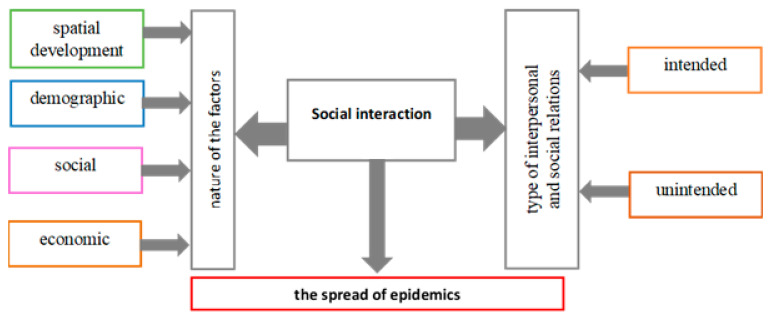
Research process pattern for determining the classification of areas with different levels of vulnerability to epidemic threats (KLE). Source: own elaboration. “Spatial development” refers to the factors related to the management of space by humans (e.g., through buildings).

**Figure 3 ijerph-19-13977-f003:**
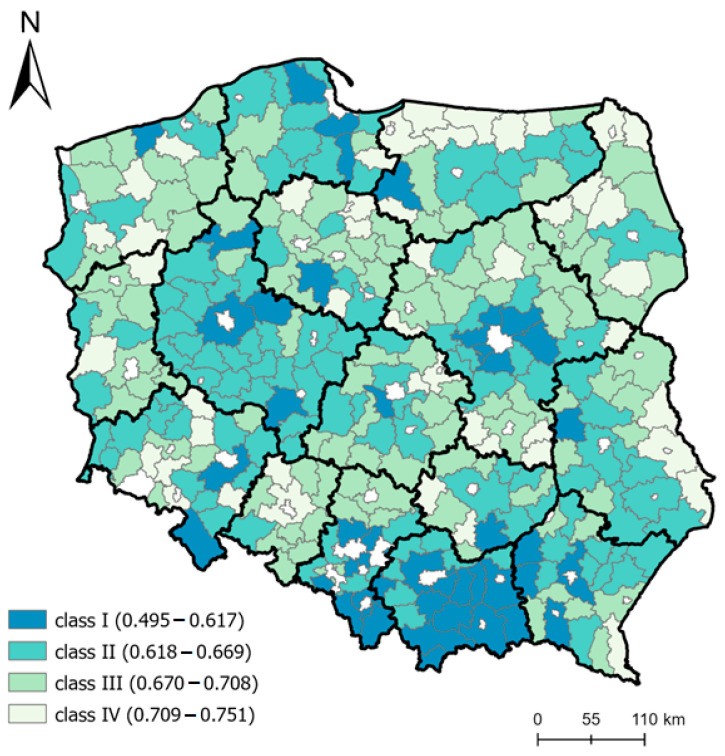
Model for classifying areas with different levels of vulnerability to epidemic threats. Source: own elaboration.

**Figure 4 ijerph-19-13977-f004:**
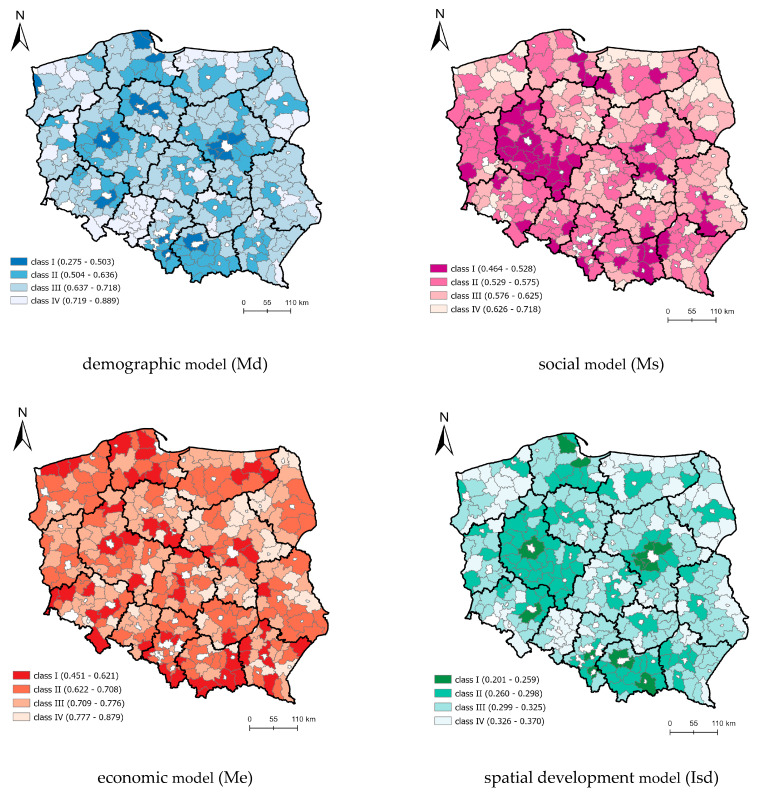
Spatial delimitation of component models due to the nature of factors. Source: own elaboration.

**Figure 5 ijerph-19-13977-f005:**
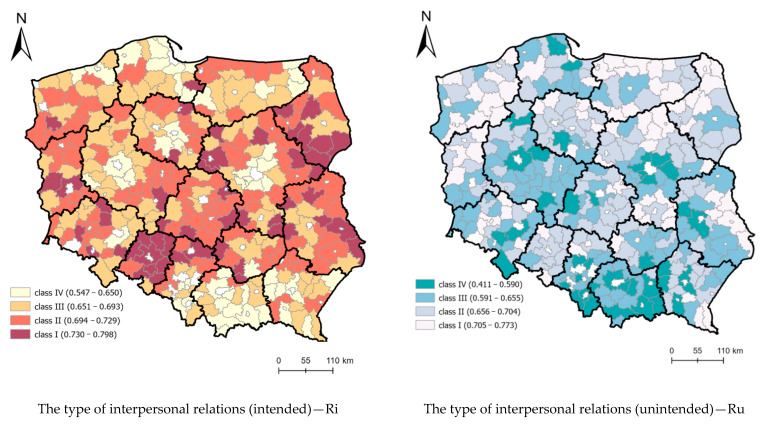
Spatial delimitation of intended and unintended interpersonal relations. Source: own elaboration.

**Figure 6 ijerph-19-13977-f006:**
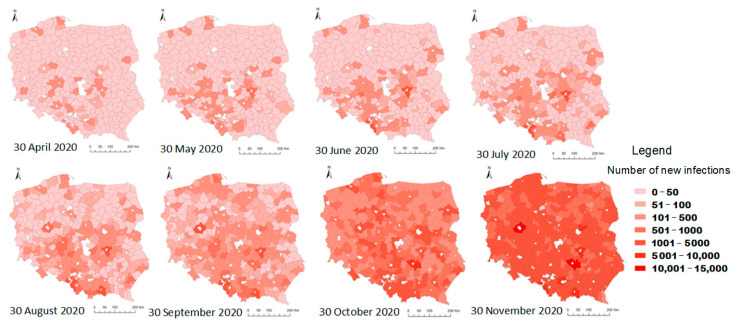
Spread of COVID-19 according to cases registered in Poland from 30 April to 30 November 2020—the first pandemic wave. Source: own elaboration.

**Figure 7 ijerph-19-13977-f007:**
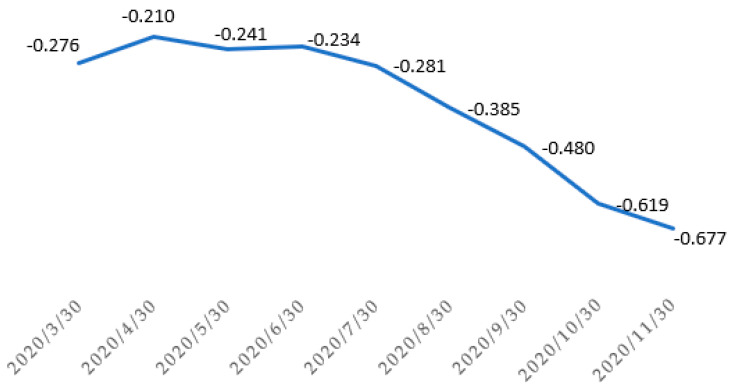
Relationship between the KLE model and the spread of COVID-19 (30.04–30.11.2020). Source: own elaboration.

**Figure 8 ijerph-19-13977-f008:**
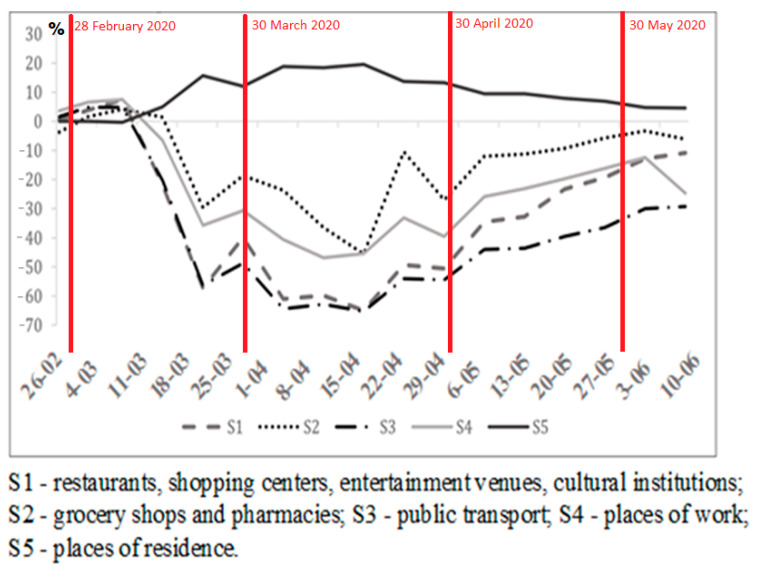
Average weekly changes in mobility in Poland from 20 February 2020 to 14 June 2020. Source: own elaboration based on [76].

**Figure 9 ijerph-19-13977-f009:**
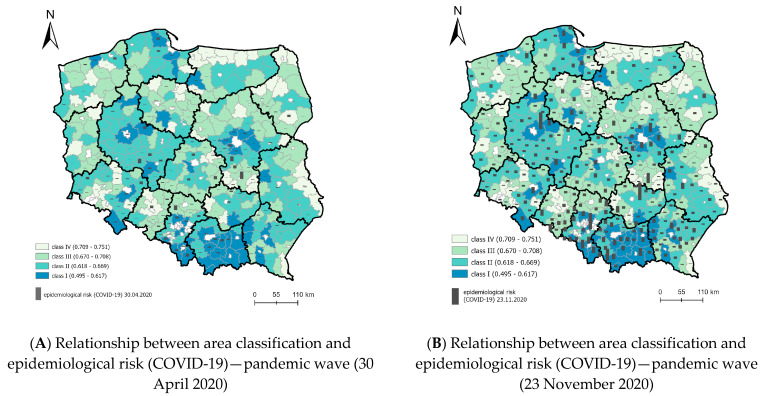
Dependency/relationship between the classification of areas (summary synthetic index WO) and epidemiological risk (number of cases) (COVID-19)—first pandemic wave. Source: own elaboration.

**Figure 10 ijerph-19-13977-f010:**
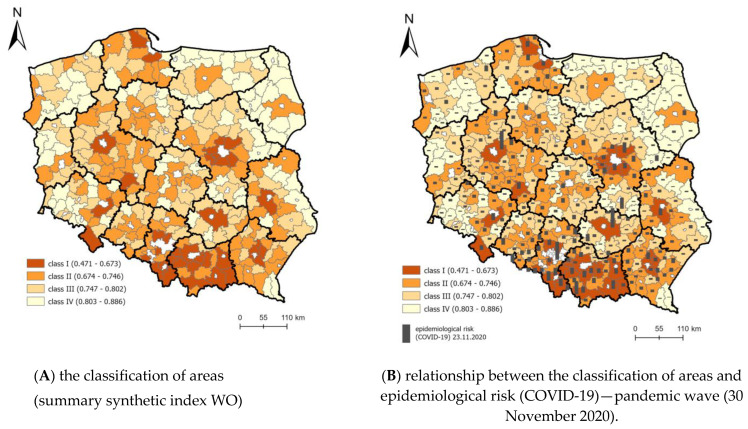
Dependency/relationship between the classification of areas (summary synthetic index WO) and epidemiological risk (number of cases) (COVID-19)—pandemic wave (30 November 2020). Source: own elaboration.

**Figure 11 ijerph-19-13977-f011:**
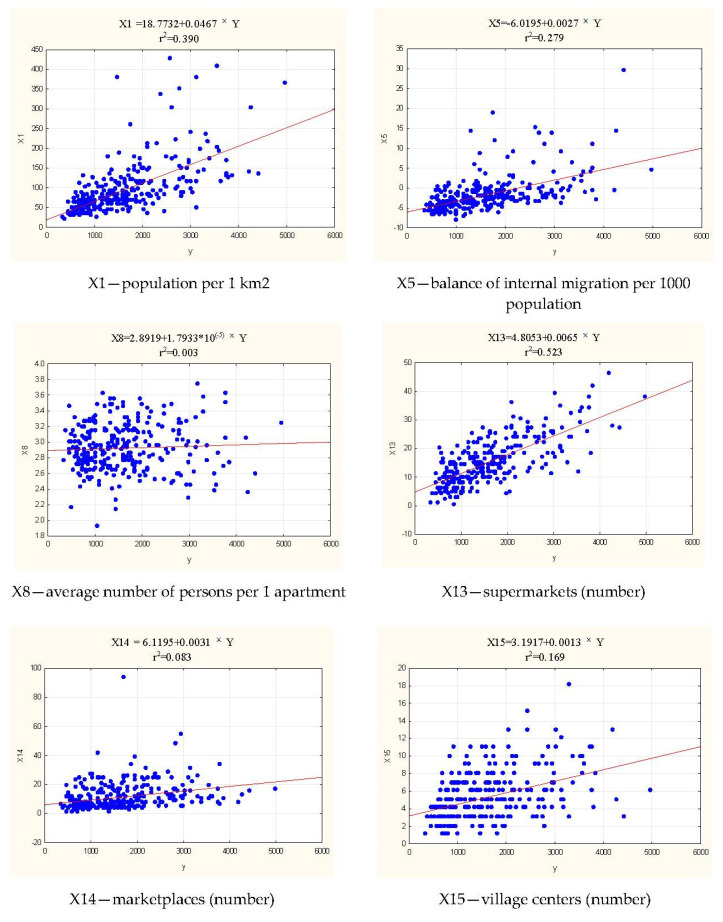
Relationship between the model of vulnerability to epidemic risks (KLE(pop)) and 11 accepted variables. Source: own study.

**Table 2 ijerph-19-13977-t002:** Social interactions according to nature of factors and type of relations (Pearson’s correlation).

Factor	Social Interactions					
	Nature of the Factors				Type of Interpersonal and Social Relations	
	Md	Ms	Me	Isd	Ri	Ru
	Correlation Means Coefficients Significant with *p* < 0.500, n = 293					
30 March 2020	−0.298	−0.126	−0.165	−0.292	−0.109	−0.315
30 April 2020	−0.176	−0.091	−0.163	−0.180	−0.046	−0.241
30 May 2020	−0.175	−0.121	−0.204	−0.192	−0.079	−0.277
30 June 2020	−0.171	−0.094	−0.196	−0.177	−0.098	−0.270
30 July 2020	−0.202	−0.122	−0.211	−0.214	−0.152	−0.304
30 August 2020	−0.286	−0.170	−0.257	−0.301	−0.224	−0.401
30 September.2020	−0.372	−0.209	−0.304	−0.387	−0.303	−0.484
30 October 2020	−0.454	−0.256	−0.413	−0.472	−0.386	−0.607
30 November 2020	−0.530	−0.288	−0.484	−0.547	−0.415	−0.683

**Table 3 ijerph-19-13977-t003:** Modeling results for determining variables.

Independent Variable	Coefficient b	Sdev.	t-Student	*p*	Nature of the Factors
X1—population per 1 km^2^	2.31	0.64	3.61	0.0004	D (i,u)
X5—balance of internal migration per 1000 population	43.12	6.92	6.23	0.0001	D (i,u)
X8—average number of persons per 1 apartment	416.43	96.55	4.31	0.0001	S (i)
X13—supermarkets (number)	40.30	4.22	9.55	0.0001	E (u)
X14—marketplaces (number)	61.61	10.14	6.08	0.0001	E (u)
X15—village centers (number)	8.33	4.06	2.05	0.0412	S (i)
X16—events (number)	0.21	0.10	2.11	0.0360	S (i)
X18—number of people per facility (cultural center, community center, club, community hall)	0.01	0.001	2.03	0.0436	S (i)
X21—population density of built-up and urbanised areas (person/km^2^)	0.20	0.06	3.22	0.0014	L (u)
X28—residents of nursing homes per 1000 inhabitants	23.08	11.10	2.08	0.0385	S (i)
X29—children in pre-school education establishments per 1000 children aged 3–5 years	1.10	0.34	3.23	0.0014	S (i)
constant	810.33	354.02	2.90	0.0001	
R	0.880				
Adjusted R^2^	0.766				
Number of observations	298				
F(11,298)	86.138				

Source: Own study on Statistica 13.1.

## Data Availability

Not applicable.

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
