# Peer review of "Social Factors as Major Determinants of Rural Development Variation for Predicting Epidemic Vulnerability: A Lesson for the Future"

_ijerph, 2022, doi:10.3390/ijerph192113977_

Round 1

Reviewer 1 Report

The research content is complete and the data is detailed in this paper, and the study has implications for the analysis of factors associated with pandemic infectious diseases. However, there are points for improvement in the paper in general, and the following suggestions are provided for the authors:

 1. The case of “COVID-19” should be consistent throughout the manuscript.

2. The data in Table 1 are for 2014, do they differ significantly from the corresponding data for 2019 and beyond?

3. Table 1 contains up to 33 variables, is there any consideration of correlation or multicollinearity between variables?

4. Line 210, line 347, etc. Suggest a canonical date format.

5. Line 221 "Etap 4" is it "stage"?

6. Not see the first level heading "3".

7. 3.1 is a bit abrupt and it is suggested to discuss or optimize its logical relationship in the paper in the context of Table 1.

8. 3.2.2 is not useful if we only analyze the spatial and temporal distribution of cases, and it is suggested to analyze district indicators, important factors affecting the number of cases, and spatial and temporal trends. In the case of this paper, it can be integrated with the section "4. Discussion".

9. Line 501 Figure 8 should be Figure 11.

10. Section 4 is labeled 4.1 only, no 4.2, 4.2....

11. The results in Table 3 seem to conflict with those in Figure 11; how appropriate is this method of selecting important variables and then fitting and analyzing the coefficients separately?

12. The statement "Local socio-economic conditions contribute to considerable variations in the spread of diseases" is questionable, as the paper focuses on the correlation or association between district economic and social indicators and cases; the causal relationship between district economic and social indicators, and the number of cases needs to be further determined and analyzed; in addition, this paper maybe not a study of the spread and diffusion of COVID-19.

13. The format of the references should be consistent and it is recommended to double-check.

Author Response

Authors would like to thank Reviewer 1 for such a thorough review and constructive remarks.

  1. Reviewer: „The case of “COVID-19” should be consistent throughout the manuscript”

Authors: In the opinion of the authors, the reviewer rightly pointed out that the notation "COVID -19" was not consistent in the text, which has been corrected (line 37, 451)

  1. Reviewer: „The data in Table 1 are for 2014, do they differ significantly from the corresponding data for 2019 and beyond?

Authors: In determining the proposed indicators we have used 2014 site area data for the feature.  There is currently no more recent data for LAU-1 level in the statistics. However, this feature does not change significantly over time and we have therefore used this data.

  1. Reviewer: „Table 1 contains up to 33 variables, is there any consideration of correlation or multicollinearity between variables?

Authors: For the analysis, we chose variables that were not correlated with each other. We used Pearson's correlation coefficient in the analysis. Its critical value is above |0.6|.

  1. Line 210, line 347, etc. Suggest a canonical date format.

Authors: As suggested by the reviewer, we have corrected the date notation (2020-04-30; 2020-11-30).

  1. Reviewer: “Line 221 "Etap 4" is it "stage"?

Authors: In the work, in order to standardise the notation, we have changed the etap to stage.

  1. Reviewer; “Not see the first level heading "3".

Authors: As recommended by the reviewer, we have added level heading 3.0 Results of empirical surveys.

  1. Review: “3.1 is a bit abrupt and it is suggested to discuss or optimize its logical relationship in the paper in the context of Table 1.”

Authors: Following the reviewer's suggestion, the authors have moved section 3.1 to section 1. At the same time, we have referred to the results recorded in Table 1. We have added information on the review of 83 research papers conducted to determine the association between socio-economic and land-use factors and an area's vulnerability to epidemic hazards using Scopus, Web of Science Core Collection and Google Scholar browsers based on the following keywords: COVID-19, social factors, economic factors, space and land-use, epidemic hazards, geographical concentration. Based on this review of articles, a total of 33 factors were identified (Table 1).

  1. Reviewer: “3.2.2 is not useful if we only analyze the spatial and temporal distribution of cases, and it is suggested to analyze district indicators, important factors affecting the number of cases, and spatial and temporal trends. In the case of this paper, it can be integrated with the section "4. Discussion".

Authors: We Agree with the reviewer's comment. Section 3.3.2 has been integrated with section 4 Discussion.

  1. Reviewer: “Line 501 Figure 8 should be Figure 11”

      Authors: As recommended by the reviewer, we have corrected line 501 where figure 8 should be figure 11.

  1. Reviewer: Section 4 is labeled 4.1 only, no 4.2, 4.2....

Authors: As recommended by the reviewer, we have corrected Section 4 by adding further labels (4.1, 4.2 ...).

  1. Reviewer: “The results in Table 3 seem to conflict with those in Figure 11; how appropriate is this method of selecting important variables and then fitting and analyzing the coefficients separately?”

Authors: The method for selecting important variables was based on stepwise regression model . For easier interpretation of the relationship between the obtained variables and the explanatory variable. A linear relationship with one variable was defined. This makes interpretation easier. We have removed from the article a linear relationship with one variable.

  1. Reviewer: “The statement "Local socio-economic conditions contribute to considerable variations in the spread of diseases" is questionable, as the paper focuses on the correlation or association between district economic and social indicators and cases; the causal relationship between district economic and social indicators, and the number of cases needs to be further determined and analyzed; in addition, this paper maybe not a study of the spread and diffusion of COVID-19”.

Authors: We agree with the reviewer's comment this sentence was not written with precision, it has been removed from the text.

  1. Reviewer: “The format of the references should be consistent and it is recommended to double-check”

Authors: As recommended by the reviewer, the format of the references has been re-checked.

The authors thank you very much for your time and effort on our work. We look forward to hearing from You regarding our revised manuscript. We would be happy to answer any further questions or comments you may have.

                                                                                               Authors

Reviewer 2 Report

1) The abstract is rather weak. Please enhance it by including the general results of the research.

2) A great number of previous research is mentioned in the introduction. However, either the gap or shortcomings of the literature or practical urgency of the issue has not been mentioned anywhere in the literature review section. The authors continue to describe different studies but did not mention the gap in the literature and practice. This section must be rewritten as well.

3) It is not clear what kind of research studies impacted the development of the research hypothesis (line 82-83).

4) Expand KLE (line 86) by first mentioning and then start using abbreviation. The same is with LAU (line 109).

5) The authors should describe the remaining sections of the study in introduction.

6) It is not clear what do you mean under "spatial development" (fig. 2). Probably either a more precise word combination should be choosen or some comments could be done (line 76 or elsewhere) in respect to this kind of impacting factors.

7) Please find a more appropriate place for the section 3.1 Theoretical Framework, as the theoretical basis of research is usually placed before the description of research methods and materials.

8) Fig. 8 - please add unit of measurement 

9) Please enhance the conclusion section in line with the research questions, include more limitations and future research directions.

10) The paper seems incoherent as there are hardly any linking conjunctions, adverbs, propositions used in the whole manuscript.

Author Response

Authors would like to thank Reviewer 1 for such a thorough review and constructive remarks.

  1. Reviewer: “The abstract is rather weak. Please enhance it by including the general results of the research”.

Authors: As recommended by the reviewer, we have revised the abstract and enriched it with the results of the study (line 22-26)

  1. Reviewer: A great number of previous research is mentioned in the introduction. However, either the gap or shortcomings of the literature or practical urgency of the issue has not been mentioned anywhere in the literature review section. The authors continue to describe different studies but did not mention the gap in the literature and practice. This section must be rewritten as well.
  2.  

Authors: As recommended by the reviewer, the work highlights the need to fill the gap in literature and practice covering the problem addressed. And this work attempts to address the existing need. 

“To the authors' knowledge, this topic has not been studied in the literature, so there is a need to fill the gap in the literature and practice regarding the problem addressed. This paper is innovative for three main reasons: (1) it comprehensively identified and classified factors available from public statistics that allow the classification of areas vulnerable to epidemic threats; (2) the significance level of the proposed factors in determining the susceptibility to epidemic threats for the territory of Poland was determined, and (3) the level of susceptibility of rural areas of Poland to epidemic threats was determined.”

  1. Reviewer: “It is not clear what kind of research studies impacted the development of the research hypothesis (line 82-83)”.

Authors: The preliminary literature review found that the COVID-19 epidemic developed faster in economically highly developed areas with higher GDP [12]. Other factors correlated with the number of cases and deaths from COVID -19 are the average age, population density and the share of people employed in caring for the elderly, the percentage of school-age children and children in daycare, and the density of doctors [16]. The spread of COVID-19 has also been linked to people travelling (migrating) from Wuhan in the Hubei province [4] and migration in the country based on returning tourists, mainly from ski re-sorts [17]. Another accelerator of the pandemic was participation in festivals [17].

  1. Reviewer: “Expand KLE (line 86) by first mentioning and then start using abbreviation. The same is with LAU (line 109)”.

Authors: As recommended by the reviewer, the name (KLE; LAU) was expanded first and then the abbreviation was used.

  1. Reviewer: “The authors should describe the remaining sections of the study in introduction”.

Authors: In accordance with the reviewer's recommendations, the introduction has been supplemented with information on the other parts of the study (line 125-130)

  1. Reviewer: “It is not clear what do you mean under "spatial development" (fig. 2). Probably either a more precise word combination should be choosen or some comments could be done (line 76 or elsewhere) in respect to this kind of impacting factors.

Authors: In the authors' view, the reviewer was right to point out that the text lacks an explanation of what is meant by the term „spatial development”. This information has been added in the text (line 213-214)

  1. Reviewer: “Please find a more appropriate place for the section 3.1 Theoretical Framework, as the theoretical basis of research is usually placed before the description of research methods and materials”.

Authors: Following the reviewer's recommendation, Section 3.1, Theoretical Framework, was placed before the description of the research methods and materials.

  1. Reviewer: “Fig. 8 - please add unit of measurement”.

    Authors: As recommended by the reviewer, we have added the unit of measurement (Fig. 8).

  1. Reviewer: “Please enhance the conclusion section in line with the research questions, include more limitations and future research directions”.

Authors: In accordance with the reviewer's recommendations, the conclusions were completed in line with the research questions and, in addition, the limitations of the research were described and future research directions were indicated (line 653-660, 697-705)

  1. Reviewer: “The paper seems incoherent as there are hardly any linking conjunctions, adverbs, propositions used in the whole manuscript”.

Authors: We hope that the additions we have made, as recommended by the reviewers, have improved the consistency of our paper.

The authors thank you very much for your time and effort on our work. We look forward to hearing from You regarding our revised manuscript. We would be happy to answer any further questions or comments you may have.

                                                                                               Authors

Round 2

Reviewer 1 Report

I would like to thank the authors for their responses to the suggestions and their efforts to make the revisions. From the author's response, it seems that the author did not explain the doubts sufficiently, but only mechanically adjusted or deleted according to the suggestions, without considering in depth why or how the changes would affect the logic and content of the paper. In the content and logic of the paper, the author should be well thought out, and partial changes or deletions are bound to have an impact on the whole. Therefore, when revising, you should pay attention to the consistency of the content and logic of the paper. For example:

Simply moving “1.1. theoretical framework” to "1. Introduction" and “section 3.2.2” to "4. Discussion" is obviously inappropriate. The content needs to be integrated rather than moved.

In response to 11. The results in Table 3 seem to conflict with those in Figure 11; how appropriate is this method of selecting important variables and then fitting and analyzing the coefficients separately?, the author does not explain whether the results in the table are consistent with those shown in the figure, and after that, the author directly deleted this part which is doubtful, I wonder if the author has considered the role of this part in the paper and its relationship with the whole. If the authors follow this way of revising, it may make the paper messy and illogical.

The others are not described in detail.

Therefore, I hope that the authors will consider the reasons for the revision and evaluate the reasonableness and effectiveness of the revision, because the authors themselves are the ones who know the paper best.

Author Response

Dear Rewiever,

Thank You for Your suggestions to the article , which have significantly improved its quality.

We agree with the reviewer that our changes should be thoroughly considered. Therefore, we have re-written our article to eliminate repetition and maintain logical consistency while not losing the findings.

Rewiever: Simply moving “1.1. theoretical framework” to "1. Introduction" and “section 3.2.2” to "4. Discussion" is obviously inappropriate. The content needs to be integrated rather than moved.

Authors: At the suggestion of the reviewer, we have integrated the theoretical framework and the introduction. We have added the following quotations to the introduction :

  1. Epidemiological risk is mostly associated with infectious diseases and preventing their occurrence and spread [12]. Most dangerous diseases pestering humanity for centuries were gradually contained and in the second half of the 20th century attention was paid to the spread of infectious diseases. The cause-and-effect relationship between the frequency of social (interpersonal) relations and the intensity of infections was demonstrated [13]. More than 2000 years ago Hippocrates claimed that environmental factors could lead to diseases. However, no measures were taken until the 19th century when intensive investigation into phenomena determining the spread of diseases in human populations commenced [14]. Studies comparing prevalence rates of human diseases became an important tool for demonstrating a relationship between living conditions or environmental factors and certain diseases. This approach was mainly used in epidemiology [15], where the subject of study is the population living in a specific area at a specific time. The population structure in various areas differs and varies in time. In the epidemiological analysis behaviour, lifestyle and social relations affecting human health are essential [16]. (Line 48-62).
  2. The characteristics of a society reveal how people interact and thereby spread diseases [19]. Their significance may vary in respective countries due to tradition, mentality and lifestyle. Many cause-and-effect relationships affect interpersonal and social interaction patterns. They are inter-actions between groups or between an individual and a group, manifested as mutual in-fluence on their behaviour, sometimes referred to as social activity [20,21]. Social inter-actions result from family, social and professional ties and can be looked into from the perspective of their direction and intensity, form and structure. They can also be investigated in terms of the underlying demographic, social, and economic factors specific to a local community. The structure of social interactions relates to the configuration of possible links between entities. These can be arrangements between two persons, groups, or individuals and groups as well as associations, networks, coalitions and clusters. The direction, intensity and form of social interaction are interlinked, constituting a complex process which - due to the human factor - can take a different course even with the same participants. In daily life a person enters into social interactions intentionally – consciously seeking them and targeting their actions at other people [22,23] – or unintentionally – when in certain relations individuals or groups of people interact accidentally [24]. The first – intended – relation may relate to establishing and maintaining family or professional interactions in connection with needs, objectives and motives, with a permanent frequency. The other – unintended – relation refers to actions aiming to limit the influence of others; encounters and contacts with other people are occasional and do not aim at maintaining ties. (line 68-88).

We have removed those parts of the text which were repetitive:

Line 64-65, line 155 - 157.

Based on the reviewer's suggestion, we added section 3.2 as an integral part of the discussion (introducing linking sentences);

Line 537 - 539; Line 562 – 564.

Rewiever: “In response to “11. The results in Table 3 seem to conflict with those in Figure 11; how appropriate is this method of selecting important variables and then fitting and analyzing the coefficients separately?”, the author does not explain whether the results in the table are consistent with those shown in the figure, and after that, the author directly deleted this part which is doubtful, I wonder if the author has considered the role of this part in the paper and its relationship with the whole. If the authors follow this way of revising, it may make the paper messy and illogical”

Authors: This reviewer suggestion influenced the structuring of the research methodology. Step 6 was added to the methodology to clarify the results in Fig. 11.

  1. Change introduced (lines 377- 385)

Stage 6. Identified relationships between the attributes for the KLE (Stage 4), and improved KLE model (Stage 5). Determining a linear model with one variable.

x(1,2,..n)=b0+b1*Y

where:

X1,2,…n — attributes for the KLE

Y—improved KLE model

b0—free term

b1 — model parameter.

  1. The amendment introduced had the effect of clarifying the provisions in the discussion (lines 688-689).

Table 3 shows the results of the relationship between the dependent variable of Covid-19 infection (Y) and the independent variables from the public statistics (X1,2,3....n). Analysis was performed using the linear multivariate stepwise regression model.

Figure 11 illustrates a different (inverse) relationship. A single relationship between a variable (X1,2,3....n) and the dependent variable of infection (Y). A linear model with a single variable was used for this purpose.

The results in Table 3 show a different relationship and the analysis performed is a different method than the results in Fig. 11. Therefore, the results obtained are different.

We have clarified the entries in the methodology and discussion. As a result, we have restored the results in the article (line 678-711 and Fig. 11), which we previously removed.

We hope that the description of the adjustments made is detailed and comprehensive. Once again, thank You very much for Your help.

Authors

Reviewer 2 Report

As I can see, the authors have seriously revised the manuscript and corrected the previously identified shortcomings. I believe that in this version the manuscript can be published in the journal.

Author Response

Dear Reviewer,
Thank You very much for appreciating our work.  Your suggestions have helped to improve the quality of our article. 

Authors